

# Shape-temperature relationships of pristine ice crystals derived from polarimetric cloud radar observations during the ACCEPT campaign

**A. Myagkov, P. Seifert, U. Wandinger, J. Bühl, and R. Engelmann**

Leibniz Institute for Tropospheric Research (TROPOS), Permoserstr. 15, 04318, Leipzig, Germany

Received: 24 November 2015 – Accepted: 9 December 2015 – Published: 15 January 2016

Correspondence to: A. Myagkov (myagkov@tropos.de)

Published by Copernicus Publications on behalf of the European Geosciences Union.

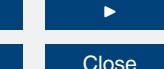
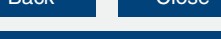
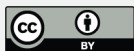

**AMTD**

doi:10.5194/amt-2015-365

**Shape-temperature relationship of pristine ice crystals**

A. Myagkov et al.

## Abstract

This paper presents first quantitative estimations of ice particle shape at the top of liquid-topped clouds. Analyzed ice particles were formed in the presence of super-cooled water and in the temperature range from −20 to −3˚C. The estimation is based on polarizability ratios of ice particles measured by a Ka-band cloud radar MIRA-35 with hybrid polarimetric configuration, manufactured by METEK GmbH. For this study, 22 cases observed during the ACCEPT (Analysis of the Composition of Clouds with Extended Polarization Techniques) field campaign were used. Polarizability ratios retrieved for cloud layers with the cloud top temperatures of ∼ −5, ∼ −8, ∼ −15, and ∼ −20˚C were 1.6, 0.9, 0.6, and 0.9, respectively. Such values correspond to prolate, quasi-isotropic, oblate, and quasi-isotropic particles, respectively. Data from a free-fall chamber were used for the comparison. A good agreement of detected shapes with well-known shape–temperature dependencies observed in laboratories was found. Polarizability ratios used for the analysis were estimated for areas located close to the cloud top where aggregation and riming processes do not strongly affect ice particles. We concluded, that in microwave scattering models ice particles detected in these areas can be assumed to have pristine shapes. It was also found that even slight variations of ambient conditions at the cloud top with temperatures warmer than ∼ −5˚C can lead to rapid changes of ice crystal shape.

## 1 Introduction

Mixed-phase clouds are a crucial component of the Earth's climate system. Their long-lasting nature impacts the radiative budget and the thermodynamic structure of the atmosphere (Sun and Shine, 1995) and microphysical processes occurring in mixed-phase clouds are the main source of precipitation (Mülmenstädt et al., 2015). Yet, current representations of mixed-phase clouds in cloud-resolving and numerical weather prediction models have significant uncertainties because numerous processes occur-

Discussion Paper | Discussion Paper | Discussion Paper | Discussion Paper |

ring in this type of clouds are understood insufficiently (Luo et al., 2008; Klein et al., 2009; Morrison et al., 2009; Bouniol et al., 2010; Delanoë et al., 2011). The uncertainties result from an incomplete understanding of the formation, microphysical evolution, and removal processes of mixed-phase clouds. The transition from the liquid phase to the ice phase and its partitioning depends strongly on the environmental temperature and the properties of the available ice nucleating particles (DeMott et al., 2010; Hoose and Möhler, 2012; Kanitz et al., 2011; Seifert et al., 2015). Temperature, pressure, and humidity at which ice crystals are formed also define their shape and apparent density which determine the sedimentation velocity (Fukuta and Takahashi, 1999; Bailey and Hallett, 2009). The vertical and dynamical structure of mixed-phase clouds furthermore determines the type and intensity of ice multiplication processes whose occurrence are a prerequisite for the formation of intensive precipitation (Hallett, 1974; Rangno and Hobbs, 2005; Seifert and Beheng, 2006).

Ground-based remote sensing has shown a large potential for improving the understanding of the life cycle of mixed-phase clouds (Hogan et al., 2003; Ansmann et al., 2009; De Boer et al., 2009; Delanoë and Hogan, 2010; Kanitz et al., 2011; Westbrook and Illingworth, 2013). Even though microphysical retrieval techniques based on ground-based remote observations are a valuable source of information for investigation of mixed-phase clouds, further developments are required in order to increase the accuracy of these retrievals. Important but yet barely explored parameters are the shape and apparent density of an ice crystal population. Estimates of ice mass, area, or number concentration require accurate knowledge of particle shape (Westbrook and Heymsfield, 2011; Delanoë et al., 2014).

From the remote-sensing perspective, mixed-phase clouds with a single supercooled liquid layer at the top and ice virgae below are of special interest (Wang et al., 2004; Smith et al., 2009). Further, we denote such clouds as single-layer clouds. Single-layer clouds have less complex microphysical and dynamical properties (Fleishauer et al., 2002; Ansmann et al., 2009; Zhang et al., 2012) compared to convective cloud systems where more than 25 different transfer processes may take place (Seifert and Beheng,

**AMTD**

doi:10.5194/amt-2015-365

**Shape-temperature relationship of pristine ice crystals**

A. Myagkov et al.

2006; Tao and Moncrieff, 2009). Thus, studying ice formation in single-layer clouds is a key to obtain a comprehensive picture of the formation of pristine ice crystals under ambient conditions.

Long-term polarimetric lidar observations showed that the majority of ice crystals in mixed-phase clouds are formed heterogeneously within a supercooled liquid layer (De Boer et al., 2011). Westbrook and Illingworth (2011) reported that about 95 % of ice particles at temperatures warmer than −20°C originated from liquid water particles. Thus, ambient conditions at the top of single-layer clouds play a crucial role in the formation of ice particles. Microphysical properties of pristine ice crystals under controlled ambient conditions have been intensively investigated in laboratories. In situ measurements in free fall chambers provide information about mass, size, shape, apparent density, and fall velocity of ice crystals at different stages of their development (Fukuta, 1969; Takahashi et al., 1991; Fukuta and Takahashi, 1999; Takahashi, 2014). Such studies provide extremely accurate information that can be used for the interpretation of remote observations and validation of retrieval techniques.

Radar polarimetry is known to be a powerful tool for the classification of microphysical properties of hydrometeors such as ice crystals under ambient conditions. In recent publications of Bühl et al. (2016) and Oue et al. (2015) vertically pointed cloud radars with linear depolarization (LDR) mode were used for qualitative discrimination between columnar-shaped ice particles and ones of other types. Nevertheless, quantitative shape estimations in LDR-mode are limited by strong dependence of polarimetric observations on canting angles of cloud particles (Matrosov et al., 2001). Melnikov and Straka (2013) proposed an algorithm for the estimation of shape and orientation of particles based on differential reflectivity $Z_{DR}$ and correlation coefficient $\rho_{HV}$ from a polarimetric weather radar with hybrid mode. This mode implies a simultaneous transmission of horizontally and vertically polarized components of the electromagnetic wave and simultaneous reception of signals in the horizontal and vertical channels. The differential reflectivity and the correlation coefficient are sensitive to shape, orientation, and dielectric properties of scatterers (Straka et al., 2000). For the estimation the authors

**AMTD**

doi:10.5194/amt-2015-365

**Shape-temperature relationship of pristine ice crystals**

A. Myagkov et al.

**AMTD**

doi:10.5194/amt-2015-365

**Shape-temperature
relationship of
pristine ice crystals**

A. Myagkov et al.

approximated scattering properties of columnar-shaped and plate-wise ice particles by prolate and oblate spheroids, respectively. Dielectric properties of ice particles depend on apparent ice density (Oguchi, 1983), which characterizes the ratio of air and ice within the approximating spheroidal particle. Observational weather radars are often operating nearly horizontally (at low elevation angles of the antenna). The authors claimed that under such conditions prolate and oblate particles can have similar polarimetric signatures and can not be reliably distinguished. As the result the proposed method is only applicable for data with high values of $Z_{DR}$ which can be only induced by strongly oblate particles and which, therefore, can be undoubtedly separated from prolate particles. In contrast to weather radars, cloud radars are often operated in modes with scanning in azimuth and elevation directions (Kollias et al., 2014; Lamer et al., 2014; Ewald et al., 2015). Matrosov et al. (2012) showed observational evidence that elevation dependencies of polarimetric variables can be efficiently used to discriminate between different types of ice particles.

In a previous study, Myagkov et al. (2015a) combined the two mentioned approaches into an algorithm for a quantitative characterization of shapes and orientations of ice particles based on polarimetric observations from a newly developed 35 GHz cloud radar with hybrid polarimetric configuration. This mode is widely used in polarimetric weather radars but is rarely implemented in cloud radars. In hybrid mode radars measure the differential reflectivity $Z_{DR}$ and the correlation coefficient $\rho_{HV}$. Ice particles of different types have distinct polarimetric scattering signatures that are utilized in the retrieval. For instance, differences in modeled polarimetric variables for oblate, prolate, and spherical solid ice particles are shown in Fig. 1. The algorithm contains two steps. First, using elevation dependencies of $Z_{DR}$ and $\rho_{HV}$ ice particles are classified as either oblate or prolate. Second, at a certain altitude polarizability ratio $\rho_e$ and degree of orientation $\rho_a$ are estimated from $Z_{DR}$ and $\rho_{HV}$ for every elevation angle in range from 30 to 60°. Note, that 90° elevation corresponds to the zenith direction. For every altitude mean and standard deviation of $\rho_e$ and $\rho_a$ are calculated. The polarizability ratio $\rho_e$ depends on shapes and dielectric properties of scatterers, while the degree of orienta-

tion $\rho_a$ characterizes the width of the canting angle distribution. It is assumed that the mean orientation of ice particles is horizontal. The algorithm was applied to a complex cloud system observed during the ACCEPT (Analysis of the Composition of Clouds with Extended Polarization Techniques) field campaign in Cabauw, the Netherlands.

Even though the results of the retrieval do not contradict other studies, the additional validation of the retrieved parameters would be of benefit. Note, that in this paper we only analyze polarizability ratios $\rho_e$.

During the ACCEPT campaign the performance of the hybrid-mode cloud radar was evaluated, but in-situ observations were not available. Nevertheless a number of active and passive remote sensors that were deployed during the campaign can be used for the determination of ambient conditions under which ice crystals were formed. Within this study, we use such information to indirectly validate the retrieval presented in Myagkov et al. (2015a) by comparing the obtained microphysical properties of ice crystals with those grown in a wind tunnel under laboratory conditions, and present an analysis of 22 case studies from the ACCEPT campaign.

The paper is organized as follows. Section 2 describes the instruments used for this study and the data set. Five case studies are presented in detail in Sect. 3. The results of the analysis of the 22 cases and their comparison with laboratory studies are presented in Sect. 4. Summary and conclusion are given in Sect. 5.

## 2    Instrumentation and data set

The ACCEPT measurement campaign was initiated by the Leibniz Institute for Tropospheric Research (TROPOS), Leipzig, Germany, the Technical University of Delft, the Netherlands, and METEK GmbH, Elmshorn, Germany within the ITaRS (Initial Training for atmospheric Remote Sensing) project. The major goal of the campaign was to evaluate the polarimetric capabilities of a newly developed polarimetric cloud radar of type MIRA-35 and to estimate the potential of its implementation into existing observational stations. The cloud radar had the hybrid polarimetric configuration. A detailed descrip-

**AMTD**

doi:10.5194/amt-2015-365

**Shape-temperature relationship of pristine ice crystals**

A. Myagkov et al.

tion can be found in Myagkov et al. (2015a). In the following, we denote this radar as the hybrid mode MIRA-35.

The campaign took place from 7 October to 17 November 2014 at the Cabauw Experimental Site for Atmospheric Research (CESAR), located in the Netherlands (51.971° N, 4.927° E). The CESAR observatory operated by Royal Netherlands Meteorological Institute (KNMI) is well equipped with a broad range of atmospheric remote-sensing instruments. In addition to the instruments available at CESAR and the hybrid mode MIRA-35, which was rented from METEK GmbH, the main instruments of Leipzig Aerosol and Cloud Remote Observations System (LACROS; Bühl et al., 2013) were brought to Cabauw. In Table 1 the instruments that were used for this particular study are presented.

As mentioned before, the shape retrieval of Myagkov et al. (2015a) requires information about the elevation dependencies of the polarimetric variables differential reflectivity $Z_{DR}$ and correlation coefficient $\rho_{HV}$. Therefore, we installed the hybrid mode MIRA-35 into the scanning unit of the LACROS container. An implemented scanning cycle took 15 min and included two elevation (range-height-indicator, RHI) scans from 30 to 150°. Note, that 90° elevation corresponds to the zenith direction. The elevation scans were performed perpendicular in azimuthal direction with $0.5° \, s^{-1}$ angular velocity. For an accurate determination of the polarimetric variables it is required to correct for antenna coupling effects. Within every scanning cycle the radar was pointed vertically for several minutes. This allowed us to use occasional light rain events for calibration of polarimetric variables as explained in Myagkov et al. (2015a, b).

During the campaign a second cloud radar MIRA-35 was operated as well. The radar is owned by TROPOS and has the conventional linear depolarization configuration. Further this radar is denoted as LDR MIRA-35. The radar was unmounted from the scanning unit of the LACROS container and installed into a trailer without scanning unit and, thus, was pointed vertically. In order to avoid interference between the two cloud radars, their operation frequencies were set to differ by approximately 300 MHz (35.17 GHz vs 35.5 GHz for hybrid-mode and LDR-mode MIRA-35, respectively). Ad-

Discussion Paper | Discussion Paper | Discussion Paper | Discussion Paper |

**AMTD**

doi:10.5194/amt-2015-365

**Shape-temperature relationship of pristine ice crystals**

A. Myagkov et al.

ditionally, the trailer was placed about 30 m away from the LACROS container to avoid any near-field interference. In this study, we use data from LDR MIRA-35 to estimate the temporal and spatial dimensions of the observed cloud systems.

When both liquid water droplets and ice crystals are present in a volume, cloud radar alone can hardly detect the liquid-water signatures. It is, however, well known that polarimetric lidars are powerful tools for the detection of supercooled liquid particles within mixed-phase clouds (Schotland et al., 1971; Seifert et al., 2010). We employed the multiwavelength Raman lidar Polly[XT] for this purpose. The lidar was set up near the Cabauw meteorological tower which is located about 300 m north from the measurement site where most of the other instruments were operated. In order to avoid specular reflection from horizontally aligned planar surfaces of ice crystals the laser beam of Polly[XT] was pointed to 5° off-zenith (85° elevation). The container with the lidar was oriented in such a way that the beam was above the radar site at about 4 km height.

Temperature is the main parameter controlling the efficiency of heterogeneous nucleation of ice crystals (DeMott et al., 2015) and it is lowest at cloud top. Therefore, in this study cloud top temperature is used as the reference parameter when crystal properties are investigated. In order to retrieve the temperature at the cloud tops, we used, in order of priority, either locally launched radiosondes, the microwave radiometer HATPRO, or assimilated meteorological datasets. Radiosondes of type Vaisala RS-92 were occasionally launched on-site. If no local radiosonde information was available, radiosonde data from the 00:00 UTC launch at De Bilt (WMO code: 06260; 20 km northeast of CESAR site) was used in case the cloud was observed around 00:00 UTC. If no recent radiosonde ascent was available, temperature data of HATPRO was used in non-precipitating conditions. Finally, for precipitating cases, when the operation of the microwave radiometer is hampered, we used temperature profiles from the GDAS1 dataset (available at: http://ready.arl.noaa.gov/gdas1.php) provided by the Global Data Assimilation System (GDAS, Kanamitsu, 1989) operated by the U.S. National Center for Environmental Prediction (NCEP).

# AMTD

doi:10.5194/amt-2015-365

## Shape-temperature relationship of pristine ice crystals

A. Myagkov et al.

For the analysis we have manually chosen 22 cases of mid-level mixed-phase clouds meeting the following criteria:

1. The hybrid-mode MIRA-35 was operating and the measured signal-to-noise ratio was high enough to retrieve calibrated polarimetric variables according to Myagkov et al. (2015a).

2. The investigated cloud layer did not experience seeding from upper clouds.

3. The calibrated polarimetric variables were available for more than 50 % of data points in elevation dimension within a half-scan of the hybrid-mode MIRA-35. This is a basic criteria for the horizontal homogeneity of the analyzed cloud layer. Note, that the cloud spatial homogeneity is not a major assumption of the retrieval algorithm. Instead, it is assumed that ice particles present at the same altitude (same ambient conditions) have the same shape, even if the cloud is not spatially homogeneous (Myagkov et al., 2015a).

4. Cloud top temperatures were in the range from $\approx -20$ to $\approx -0\,^{\circ}\mathrm{C}$. Ice crystals formed at such temperatures under water saturation conditions have a clear primary shape (Bailey and Hallett, 2009). At lower temperatures ice particles can have a variety of shapes at a certain temperature (Bailey and Hallett, 2004), which can significantly influence the shape retrieval.

5. For non-precipitating cases lidar data should be available.

6. At temperatures above $-5\,^{\circ}\mathrm{C}$ the presence of ice crystals should be confirmed. For cases without liquid precipitation ice virgae produce strong volume depolarization ($> 0.2$) of the lidar signal. For precipitating cases a melting layer is an indicator of ice presence. If none of the two checks was positive, the layer was excluded from further analysis.

In the ideal case, analyzed cloud layers should not produce precipitation to permit the usage of lidar and radiometer data. Nevertheless, most of the clouds with cloud

Discussion Paper | Discussion Paper | Discussion Paper | Discussion Paper |

**AMTD**

doi:10.5194/amt-2015-365

**Shape-temperature relationship of pristine ice crystals**

A. Myagkov et al.

top temperatures warmer than −5˚C that fulfilled the requirements 1 and 2 produced precipitation.

## 3 Examples of the shape retrieval

In this section we show five examples of the shape estimation retrieval of Myagkov et al. (2015a) based on elevation scans of differential reflectivity and correlation coefficient. As case studies we chose mixed-phase clouds with different cloud top temperatures that were observed during the ACCEPT campaign. Several types of formed ice crystals were clearly identified by the hybrid-mode MIRA-35. The fifth case study indicates that slight variations of cloud top altitude can lead to changes from oblate to prolate shape.

### 3.1 Case 1

Figure 2a and b represents height-time cross-sections of the equivalent radar reflectivity $Z_e$ and LDR measured by LDR MIRA-35. These parameters were calculated using the total powers of the received signal in the co- and cross-polarized channels. A cloud system observed on 12 October 2014, 15:00–16:00 UTC is depicted. The radar observed a cloud layer with the top at around 5.2 km height. From 15:32 UTC the cloud layer was influenced by a higher-level cloud with a top height at 6 km height. For the analysis we thus chose the time period 15:16–15:20 UTC when the high-level cloud did not cause any seeding effects to the lower layer. Within the chosen period the SNR was high enough to apply the shape retrieval algorithm. On the other hand, ice development in this period was not as intensive as the one starting at 15:20 UTC, which is confirmed by about 10 dB lower values of $Z_e$. Figure 2b shows that in the cloud layer ice particles did not produce depolarization. Observed values of LDR are very close to the minimum observable LDR of −31 dB. In Fig. 2c and d the attenuated backscatter coefficient and the volume linear depolarization ratio measured by Polly$^{XT}$ are presented, respectively. A single liquid layer indicated by increased values of the backscatter coefficient can

**AMTD**

doi:10.5194/amt-2015-365

Shape-temperature relationship of pristine ice crystals

A. Myagkov et al.

be clearly seen at the top of the cloud layer. Low values of volume depolarization ratio within the liquid layer are caused by the spherical shape of supercooled water drops. It is noticeable, that values of volume depolarization ratio were also low in the ice virga. The reason for this behavior is unclear, considering that Polly$^{XT}$ is pointed 5° off-zenith to prevent the influence of specular reflection at planar planes of horizontally aligned crystals. It may thus be a distinct microphysical feature of the ice crystals formed at the given temperature.

A photograph of the analyzed cloud layer is shown in Fig. 2g. It can be seen that the cloud is spatially homogeneous. Figure 2e and f depicts the differential reflectivity $Z_{DR}$ and the correlation coefficient $\rho_{HV}$ measured by the hybrid MIRA-35, which were calculated for the spectrum peaks. Note that we plot the RHI scans of $Z_{DR}$ and $\rho_{HV}$ uncorrected for the polarization coupling to make figures more illustrative. After the correction the amount of data points is much lower. Nevertheless, for the shape retrieval algorithm shown below we use the corrected values. Strong elevation dependencies can be seen in $Z_{DR}$ and $\rho_{HV}$. At 90° elevation, the differential reflectivity is almost zero whereas it reaches values of 3 dB at lower elevations. The correlation coefficient has values close to 1 in zenith direction while its values at lower elevations reach 0.98.

Using the corrected values of $Z_{DR}$ and $\rho_{HV}$ we retrieved polarizability ratios separately for the left and right half scans. For the antenna coupling correction we used vertical measurements of rain from 12 October 2014, 19:00–20:00 UTC. Results of the retrievals are given in Fig. 2h and i. The value of the polarizability ratio close to the cloud top is of special interest, because there ice particles should be least influenced by processes such as aggregation and riming which would lead to a deviation of the crystal shape from its pristine state. Unfortunately, often the SNR at the cloud top is too low to apply the retrieval. The retrieved values of the polarizability ratio closest to the cloud top is $0.62 \pm 0.09$, which corresponds to oblate spheroids. The distance from the cloud top is about 150 m. Figure 2j represents a temperature profile retrieved from the microwave radiometer HATPRO. It can be seen that at the cloud top, where ice crystals are formed, the temperature was around −14.9 °C.

**AMTD**

doi:10.5194/amt-2015-365

**Shape-temperature relationship of pristine ice crystals**

A. Myagkov et al.

Discussion Paper | Discussion Paper | Discussion Paper | Discussion Paper |

Discussion Paper | Discussion Paper | Discussion Paper | Discussion Paper |

**AMTD**

doi:10.5194/amt-2015-365

**Shape-temperature relationship of pristine ice crystals**

A. Myagkov et al.

## 3.2 Case 2

An analysis of a mid-level mixed-phase cloud observed on 18 October 2014, 01:00–02:00 UTC is given in Fig. 3. The time period chosen for the retrieval is 01:23–01:27 UTC. The cloud top estimated from equivalent radar reflectivity measured by

5  LDR MIRA-35 (Fig. 3a) was at around 5 km height. The thickness of the cloud layer exceeded 1.2 km. Reflectivity values reached values as high as 10 dBZ that indicate the presence of large ice particles. LDR values registered by LDR MIRA-35 for the analyzed period were mostly low even though areas with increased LDR (up to −17 dB) can be seen occasionally at 4–4.5 km height. At the top of the cloud a liquid layer char-

10  acterized by high attenuated backscatter coefficients and low volume depolarization ratios (Fig. 3c and d, respectively) is visible. It should be noted, that in the second half of the analyzed period the lidar also detected an internal liquid layer at 4.5 km height. In contrast to case 1, where low volume depolarization ratios were observed with Polly$^{XT}$, the ice virgae observed in the case 2 produced volume depolarization ratios exceeding

15  0.5.

   Elevation scans of the differential reflectivity and the correlation coefficient depicted in Fig. 3e and f, respectively, show that the cloud was spatially inhomogeneous. Within the right-half scan only slight angular changes in $Z_{DR}$ and $\rho_{HV}$ are visible. In the left-half scan high $Z_{DR}$ and low $\rho_{HV}$ values were observed at lower elevations at 4.5 km height

20  at which the lidar detected the internal liquid layer. For the correction of the polarization coupling we used vertical measurements in a short precipitation event on 18 October 2014, 2:30–2:50 UTC. The results of the shape retrieval are shown in Fig. 3g and h. It can be seen that due to the spatial inhomogeneity the calculated polarizability ratios are available only for a limited number of range bins. Nevertheless, the results show that in

25  the left-half scan prolate particles characterized by the polarizability ratio of 1.52 ± 0.2 were detected. In the right-half scan the polarizability ratios closest to the cloud top were 1.07±0.1. Such values correspond to particles with quasi-spherical shapes and/or low apparent ice density. Further we denote such particles as quasi-isotropic as they

**AMTD**

doi:10.5194/amt-2015-365

**Shape-temperature relationship of pristine ice crystals**

A. Myagkov et al.

do not change the polarization of the scattered wave significantly. The distances from the liquid layers where prolate and quasi-isotropic particles formed, were about 0.4 and 0.8 km, respectively. Figure 3i shows that temperatures at liquid-layer heights were −6.1 and −9.3°C, respectively. We point out that the coexistence of different types of particles can lead to misclassification of prolate and oblate particles. The spectra peaks at different elevations can be dominated by different particles. One of the ways to avoid this influence is a combined Doppler-polarimetric analysis similar to the one given by Oue et al. (2015). After the separation of spectral modes the retrieval algorithm can be applied to each mode separately. In this paper we do not provide such an analysis.

## 3.3  Case 3

In Fig. 4 a residual part of a mixed-phase cloud system observed on 20 October 2014, 18:00–19:00 UTC, is shown. We consider a cloud layer with the cloud top at around 3.6 km height in the time period from 18:16–18:20 UTC (Fig. 4a). In the chosen area the cloud layer was about 1 km thick and the radar reflectivity reached −10 dBZ. The cloud layer did not experience seeding from the higher cloud layer. High values of LDR (Fig. 4b) that reached up to −15 dB indicated the presence of strongly non-spherical scatterers. In Fig. 4c and d enhanced values of the attenuated backscatter coefficient and low volume depolarization ratios at the top of the cloud layer indicated the presence of a single supercooled liquid layer. The average volume depolarization ratio in the ice virga was ∼ 0.3.

Figure 4e and f shows that the cloud layer was spatially homogeneous. Strong angular dependencies in $Z_{DR}$ and $\rho_{HV}$ can be clearly seen. Changes in differential reflectivity were up to 2 and 4 dB within the left- and right-half scan, respectively. It is noticeable, that the correlation coefficient $\rho_{HV}$ had its minimum in zenith-pointing direction and approached higher values at lower elevations. We showed an example in Myagkov et al. (2015a) that such signatures are specific for prolate particles. For the polarization coupling correction we used vertical observations in a light rain event on 21 October 2014, 8:00–9:00 UTC. The results of the shape retrieval are depicted in Fig. 4g and h. Re-

trieved polarizability ratios are slightly higher in the right half-scan which is caused by the observed increased values of $Z_{DR}$. The polarizability ratios closest to the cloud top are $1.5 \pm 0.16$. The distance from the cloud top is about 240 m. A temperature profile retrieved from the microwave radiometer HATPRO indicated that the temperature at the cloud top was $-6.1\,°C$ (see Fig. 4i).

### 3.4 Case 4

A complex mixed-phase cloud system observed on 10 November 2014, 02:00–03:00 UTC is presented in Fig. 5. A cloud layer with the cloud top at around 5 km height in the time period 02:53–02:57 UTC is considered for the analysis. The cloud layer was more than 2 km thick. The equivalent radar reflectivity at the cloud top did not exceed $-10$ dBZ while it reached values up to 10 dBZ towards the cloud bottom (Fig. 5a). LDR values measured by LDR MIRA-35 were about $-30$ dB (Fig. 5b). In Fig. 5c it can be seen that the laser beam often could not penetrate through the whole cloud layer because of strong attenuation. Nevertheless, some indications of liquid water at the cloud top are present. For example, two areas at 5.4 km height characterized by the low volume depolarization ratio can be identified. There is also a thick internal liquid layer at about 4 km height specified by the increased attenuated backscatter coefficient and the low volume depolarization ratio.

Figure 5e and f shows almost no angular dependencies of $Z_{DR}$ and $\rho_{HV}$. Some slight changes are visible at 4 km height where the lidar detected the internal liquid layer. For the correction of the antenna coupling we used observations during a short precipitation event on 1 November 2014 17:00–18:00 UTC. The retrieved polarizability ratios were close to 1 characterizing quasi-isotropic particles. The retrieved profiles indicated rapid changes of the polarizability ratio from 0.92 to 1.05, e.g., from 4.5 to 4.8 km height in the left-half scan. Such changes result from misclassification of prolate and oblate particles which is caused by a variability in $Z_{DR}$ and $\rho_{HV}$ due to measurement noise and/or differences in scattering properties of ice populations. Biases in polarimatric variables caused by the polarization coupling, also lead to inaccurate classification.

**AMTD**

doi:10.5194/amt-2015-365

**Shape-temperature relationship of pristine ice crystals**

A. Myagkov et al.

Discussion Paper | Discussion Paper | Discussion Paper | Discussion Paper

Discussion Paper | Discussion Paper | Discussion Paper | Discussion Paper |

Without the polarization coupling correction given by (Myagkov et al., 2015b) the misclassification for the used radar would occur in the range of polarizability ratios from 0.8 to 1.2. For the following analysis we chose the polarizability ratio of $0.92 \pm 0.08$ observed 450 m below the cloud top. The temperature measured at 5.5 km height by the radiosonde launched at midnight from De Bilt was about $-20\,°C$.

## 3.5 Case 5

Figure 6 depicts a precipitating cloud system with the cloud top located at around 2.3 km height which was observed on 7 November 2014, 20:00–21:00 UTC. The SNR in this case was not high enough to retrieve polarizability ratios, although elevation dependencies of the measured polarimetric variables allow us to classify the general shape of the observed ice particles. We analyzed two scans performed by the hybrid MIRA-35 which correspond to the time periods shown in Fig. 6a enframed by black rectangles. The time gap between these periods is 11 min. It can be seen that values of $Z_e$ in the first period do not exceed $-15$ dBZ. Ice formation was not intensive enough to produce precipitation reaching the ground. Corresponding ice particles caused low depolarization which is indicated by low LDR values of around $-30$ dB shown in Fig. 6b. In contrast, ice formation in the second time period was much more intensive. $Z_e$ values close to the top of the cloud were around $-20$ dBZ, while those observed 1 km below the cloud top exceeded 10 dBZ. Ice particles were large enough to produce liquid precipitation at the ground with 10 dBZ equivalent radar reflectivity. Ice particles in the second period were characterized by LDR values of up to $-15$ dB. The melting layer with LDR of $-12$ dB is clearly seen at around 1.4 km height.

In Fig. 6c and d range-height cross sections of SNR for the first and second time period are given, respectively. In both cases SNR are of the same order of magnitude. Elevation scans of differential reflectivity shown in Fig. 6e and f yield $Z_{DR}$ values close to 0 dB in the zenith-pointing direction, while at lower elevations $Z_{DR}$ reached 4 and 2 dB, respectively. For both cases $Z_{DR}$ had less pronounced angular dependence at 1.5 km height. This effect can be caused by aggregation as particles become more spherical

**AMTD**

doi:10.5194/amt-2015-365

**Shape-temperature relationship of pristine ice crystals**

A. Myagkov et al.

AMTD

doi:10.5194/amt-2015-365

and/or less dense. Angular dependencies of $\rho_{HV}$ at the cloud tops show a different behavior. In Fig. 6g $\rho_{HV}$ has the highest value in the zenith-pointing direction and slightly decreases at lower elevations. For the second time period $\rho_{HV}$ has its minimum value of about 0.93 at vertical pointing direction and increases up to 0.98 at lower elevations.

Observed elevation dependencies at the cloud tops indicate the presence of oblate and prolate particles for the first and the second time periods, respectively. Unfortunately, we do not have continuous temperature profiles from the microwave radiometer for this case because of precipitation. Temporal resolution of the GDAS model is 3 h and can therefore not be used to capture temperature variations within 15 min. The temperature profile given in Fig. 6i shows the cloud top temperature of −4 °C at 2.3 km height at 21:00 UTC. In Fig. 6a it can be seen that cloud top altitudes for the analyzed time periods differ by about 200 m. Also vertical variations of LDR (Fig. 6b) indicate changes of the 0 °C isotherm. Thus, temperatures at the top could be different by few degrees which can cause crucial differences in the ice particle shape. We want to emphasize that such strong indications of the presence of oblate particles at such warm temperatures were observed twice during the whole field campaign. In both cases the existence of these particles was registered for not longer than about 5 min.

## 4 Comparison with laboratory studies

The analysis shown in Sect. 3 was applied to 22 cases. The number of cases corresponding to cloud top temperature ranges from −7 to −3, −13 to −7, −17 to −13, and −25 to −17 °C are 9, 2, 9, and 2, respectively. Most of the analyzed clouds had a liquid layer at the top which was identified using the polarimetric measurements from the lidar. In the case of precipitating clouds we rely on the conclusion of Westbrook and Illingworth (2011) and De Boer et al. (2011) that 95 % of ice crystals at temperatures warmer than −25 °C are formed in presence of liquid water. For every case we chose the polarizability ratio closest to the cloud top detected by the LDR MIRA-35. The distance from the cloud top mostly does not exceed 500 m.

Discussion Paper | Discussion Paper | Discussion Paper | Discussion Paper |

For the evaluation of the measured polarizability ratios we use measurements from fall chamber studies of Takahashi et al. (1991). In order to facilitate the comparison, first the polarizability ratio needs to be derived from the laboratory measurements. Thus, we use the information about axis lengths and mass of ice particles grown at water saturation conditions in the temperature range from $-23\ldots -3\,°C$. We define the geometrical axis ratio $\rho_g$ and the apparent ice density $\rho_a$, respectively, of an ice particle as follows:

$$\rho_g = \frac{c}{a}, \tag{1}$$

$$\rho_a = \frac{8m}{3\sqrt{3}a^2 c}, \tag{2}$$

where $m$ is mass of the ice particle and $a$ and $c$ are axis lengths of the ice particle. Note that $a < c$ for a prolate particle, while $a > c$ for an oblate particle. According to Takahashi and Fukuta (1988) apparent ice density in Eq. (2) is calculated considering the ice particle as a hexagonal prism.

The dependencies of the geometrical axis ratio and apparent ice density on the temperature at which ice particles were formed are shown in Fig. 7. At temperatures as low as $-5$ and $-15\,°C$ long columns (strongly prolate) and dendrites (strongly oblate particles) were formed in the laboratory, respectively. Quasi-spherical (also known as isometric) particles were observed near $-3$, $-10$, and $-23\,°C$. These shape-temperature dependencies have been known from laboratory measurements since the 1950s (Kampe et al., 1951), even though it is not clear yet to which extent these studies are valid at ambient conditions. Considering the apparent ice density, it can be seen that the columns and the dendrites had values of $\rho_a$ down to $0.3\,g\,cm^{-3}$. At temperatures around $-12$ and $-16\,°C$ particles had the highest values of $\rho_a$ exceeding $0.8\,g\,cm^{-3}$. Ice formation in this temperature range was studied more precisely by Takahashi (2014). Quasi-spherical particles tended to have $\rho_a$ in the range of 0.6–$0.8\,g\,cm^{-3}$.

# AMTD

doi:10.5194/amt-2015-365

**Shape-temperature relationship of pristine ice crystals**

A. Myagkov et al.

It is known that the dielectric constant of ice is almost linearly dependent on apparent ice density (Oguchi, 1983). In this study, we assume the following dependence:

$$\epsilon = 2.36\rho_a + 1, \tag{3}$$

where $\rho_a$ is in g cm$^{-3}$. We used the analytical spheroidal scattering model and information about geometrical axis ratios and dielectric constants of ice crystals grown in the fall chamber to obtain their polarizability ratios $\rho_e$ (see Eqs. 39–44 in Myagkov et al., 2015a). For the polarimetric scattering model we assume that spheroids have the same volume and geometrical axis ratio as the hexagonal prisms used for the calculation of the apparent ice density in Eq. (2). In Fig. 8 values of $\rho_e$ retrieved from the laboratory studies are shown as blue filled circles. The highest $\rho_e$ values of about 1.6–1.7 were observed in the temperature range from −7 to −6 °C. Even though prolate particles with the highest geometrical axis ratio were formed at −5 °C, they had low apparent ice density and therefore their $\rho_e$ did not exceed 1.4. Polarizability ratios of 0.4 correspond to ice crystals that formed at temperatures of −12 and −16 °C, where ice crystals were found to cause apparent ice densities exceeding 0.8 g cm$^{-3}$. Dendrites that form at −15 °C had low $\rho_a$ that led to values of $\rho_e$ of 0.6. In Fig. 8 we also show polarizability ratios retrieved from the polarimetric observations of the hybrid MIRA-35 during the ACCEPT campaign (red filled dots with error bars). A good agreement between findings from the free-fall chamber and remote observations can be seen. In the temperature range of −6 to −4 °C values of $\rho_e$ retrieved close to cloud tops varied from 1.4–1.8. At temperatures of −9 to −7 °C isometric particles were detected with $\rho_e$ of 0.8–1.2. In the temperature range of −17 to −13 °C observed oblate particles mostly had $\rho_e$ of 0.4–0.6. At temperatures from −25 to −20 °C ice crystals had $\rho_e$ of 0.8–1. At the same time differences in measured and calculated polarizability ratios can be seen in Fig. 8 at temperatures of −4 to −3 °C. Such differences can be caused by uncertainties in temperature values from the GDAS1 data set. Even though the number of cases available from 6 weeks of measurements is quite low, we show that ice particles formed close to the top of mid-level mixed-phase clouds at temperatures warmer than

Discussion Paper | Discussion Paper | Discussion Paper | Discussion Paper

**AMTD**

doi:10.5194/amt-2015-365

**Shape-temperature relationship of pristine ice crystals**

A. Myagkov et al.

**AMTD**

doi:10.5194/amt-2015-365

**Shape-temperature relationship of pristine ice crystals**

A. Myagkov et al.

~ −25°C in general have a similar dependence of shape and apparent ice density on ambient temperature as the ones grown in the free-fall chamber.

## 5  Summary and outlook

Polarimetric cloud radars have a great potential for the provision of valuable informa-
tion about ice crystal microphysical properties based on remote sensing. Within this study it was demonstrated, that the scanning capabilities of modern cloud radars al-low for the classification and quantitative characterization of ice particle shape. For this purpose a 35 GHz cloud radar MIRA-35 with the hybrid polarimetric configuration was deployed during the ACCEPT measurement campaign in Cabauw, the Netherlands in October and November of 2014. The radar provided elevation dependencies of the differential reflectivity and the correlation coefficient which were used to estimate the polarizability ratios of ice crystals approximated by spheroids. The polarizability ratio thus depends on the geometrical axis ratio and the apparent ice density. The radar was collocated with a vast number of active and passive remote sensors providing continuous information about cloud geometry, ambient conditions, and the presence of supercooled liquid layers. The combined analysis of the available data allowed us to derive temperature-dependent polarizability ratios of ice crystals in the cloud-top re-gion of mixed-phase cloud layers. During the ACCEPT campaign in-situ observations were not available. Therefore, we used data about ice crystals grown in a free-fall cloud chamber to validate our retrieval indirectly. The measurements available from labora-tory studies include accurate information about axis lengths and mass of ice crystals grown at water saturation conditions. We utilized these parameters to calculate polar-izability ratios from the laboratory measurements. A comparison of polarizability ratios of ice crystals investigated in the cloud chamber and the ones observed close to the cloud tops showed a good agreement. At temperatures in the range of −6 to −4°C columnar-shaped particles with $\rho_e$ of 1.2–1.7 and 1.4–1.8 were found in laboratory studies and remote observations, respectively. Isometric particles occurred at temper-

atures near −8 and −20˚C. Oblate particles investigated in the temperature range of −17 to −13˚C, had $\rho_e$ of 0.4–0.6 both in the chamber studies and remote observations.

From the present study, we can conclude that ice particles located close to tops of mixed-phase clouds are not significantly influenced by aggregation and/or riming and can be considered as pristine in scattering models in the microwave region. In addition, the evaluation showed that also the apparent density of pristine ice crystals that formed at water saturation conditions is comparable to the laboratory measurements. This information is of special value for modeling studies for which the findings of this study are a confirmation of the validity of laboratory studies regarding the properties of pristine ice crystals at ambient conditions.

During the ACCEPT campaign only 22 well-defined cases valid for the analysis were found. Thus, much more polarimetric observations are required to collect a data set that can be used for further analysis. Nevertheless, our findings show that the agreement between laboratory studies and field observations of freshly formed, mostly pristine ice crystals is justified in general. In future studies special attention should be paid to the analysis of spectral polarimetric variables, as this can help to separate different populations of ice crystals within a cloud and, e.g., reduce classification errors. The analysis of polarimetric variables can be also successfully used for the investigation of large ice particles such as aggregates and graupel. A Ka-band polarimetric cloud radar provides a large set of variables that still have to be interpreted for the case of large ice particles. Potentially, a combination of polarimetric, Doppler, and multi-frequency analysis may yield even more information about different types of ice particles. For instance, advances of combined Doppler measurements and polarimetry are shown in Bühl et al. (2016) and Oue et al. (2015). But first, capabilities of all mentioned approaches should be further investigated.

*Acknowledgements.* The research leading to these results has received funding from the European Union Seventh Framework Programme (FP7/2007-2013): People, ITN Marie Curie Actions Programme (2012–2016) in the frame of ITaRS under grant agreement no. 289923. The ACCEPT campaign was partly funded from the European Union Seventh Framework Pro-

Discussion Paper | Discussion Paper | Discussion Paper | Discussion Paper |

**AMTD**

doi:10.5194/amt-2015-365

**Shape-temperature relationship of pristine ice crystals**

A. Myagkov et al.

gramme (FP7/2007-2013) under grant agreement no. 262254. Authors want to acknowledge METEK GmbH, particularly Matthias Bauer-Pfundstein and Alexander Partus for their assistance in maintenance of cloud radars during the ACCEPT campaign. We also highly appreciate T. Takahashi, Hokkaido University of Education, Sapporo, Japan, for providing the data from the free-fall cloud chamber. The authors also would like to acknowledge the logistic support received from KNMI and TU Delft.

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

**Table 1.** The list of used instruments.

| Instrument | Main specifications | Measurements | Reference |
|---|---|---|---|
| Cloud radar MIRA-35 | Frequency: 35.5 GHz, configuration: LDR, pointing: zenith temporal resolution: 1 s, range resolution: 30 m | Equivalent radar reflectivity factor, LDR, mean Doppler velocity, Doppler width, complete spectra | Görsdorf et al. (2015) |
| Cloud radar MIRA-35 | Frequency: 35.17 GHz, configuration: hybrid, pointing: scanning, temporal resolution: 1 s, range resolution: 30 m | Equivalent radar reflectivity factor, mean Doppler velocity, Doppler width, complete spectra, differential reflectivity, correlation coefficient, differential phase shift | Myagkov et al. (2015a) |
| Multiwavelength Raman lidar Polly XT | Wavelengths: 355, 532, 1064 nm, pointing: 5° off-zenith, temporal resolution: 30 s, range resolution: 7.5 m | Backscatter coefficient at three wavelengths, volume depolarization ratio at 532 nm | Althausen et al. (2009) |
| Microwave radiometer HATPRO | Bands: 22–31 GHz, 51–58 GHz, temporal resolution: 1 s | Brightness temperatures, temperature profile, liquid water path | Rose et al. (2005) |
| Radiosonde Vaisala RS92 | Variable resolution | Temperature, pressure, relative humidity, wind | Suortti et al. (2008) |

**Shape-temperature relationship of pristine ice crystals**

A. Myagkov et al.



Discussion Paper | Discussion Paper | Discussion Paper | Discussion Paper

**AMTD**

doi:10.5194/amt-2015-365

**Shape-temperature relationship of pristine ice crystals**

A. Myagkov et al.

Interactive Discussion

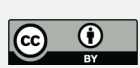

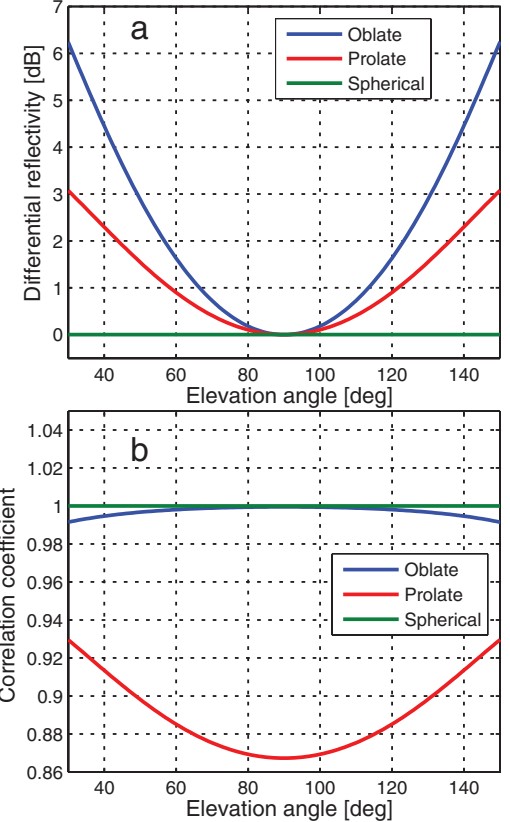

**Figure 1.** Elevation dependencies of modeled **(a)** $Z_{DR}$ and **(b)** $\rho_{HV}$ for strongly oblate (axis ratio $\ll 1$), strongly prolate (axis ratio $\gg 1$), and spherical (axis ratio of 1) solid ice particles. Particles are assumed to be oriented near horizontally with the degree of orientation of 0.99. The figure is based on look-up tables given in Myagkov et al. (2015b).

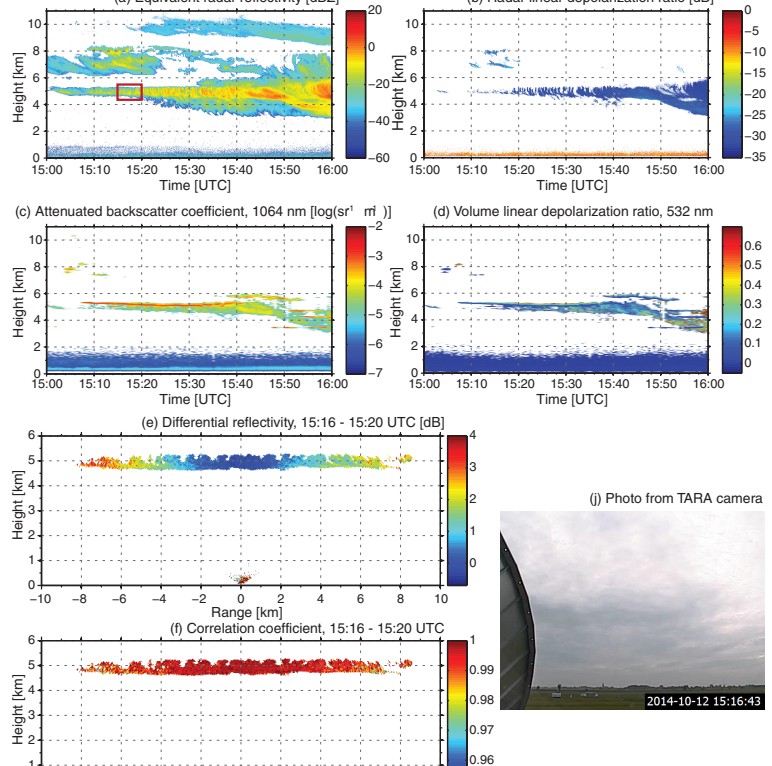

**AMTD**

doi:10.5194/amt-2015-365

**Shape-temperature relationship of pristine ice crystals**

A. Myagkov et al.

**Figure 2.** Case study of ice crystal shapes observed with hybrid-mode MIRA-35 on 12 October 2014 at Cabauw, the Netherlands. Shown are time-height cross sections of the following parameters: **(a)** equivalent radar reflectivity $Z_e$ from LDR MIRA-35; **(b)** radar linear depolarization ratio from LDR MIRA-35; **(c)** attenuated backscatter coefficient at 1064 nm from Polly-XT; and **(d)** volume linear depolarization ratio at 532 nm from Polly-XT, range-height cross sections of **(e)** differential reflectivity $Z_{DR}$ and **(f)** correlation coefficient $\rho_{HV}$ measured by hybrid MIRA-35, **(g, h)** vertical profiles of polarizability ratio for the left- and right-half scans, respectively, **(i)** vertical temperature profile from the microwave radiometer HATPRO, **(j)** photo taken by a web-camera installed at the CESAR site. The red rectangle shows the analyzed cloud layer and the time period corresponding to a full elevation scan of hybrid MIRA-35. $Z_e$ and LDR were calculated from total powers in the co- and cross-channels. $Z_{DR}$ and $\rho_{HV}$ were obtained for the spectral peaks. Note, that $Z_{DR}$ and $\rho_{HV}$ are not corrected for the antenna coupling here to make figures more illustrative. Red horizontal lines in **(h, i, j)** represents the cloud top. Vertical profiles in **(h)** and **(i)** indicated by red line correspond to mean values of polarizability ratio. Horizontal blue bars in **(h)** and **(i)** denote 2 standard deviations of polarizability ratio.

**AMTD**

doi:10.5194/amt-2015-365

**Shape-temperature relationship of pristine ice crystals**

A. Myagkov et al.

**AMTD**

doi:10.5194/amt-2015-365

**Shape-temperature relationship of pristine ice crystals**

A. Myagkov et al.

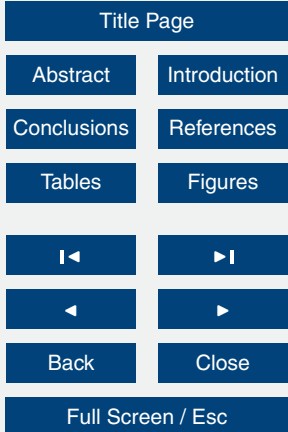

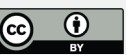

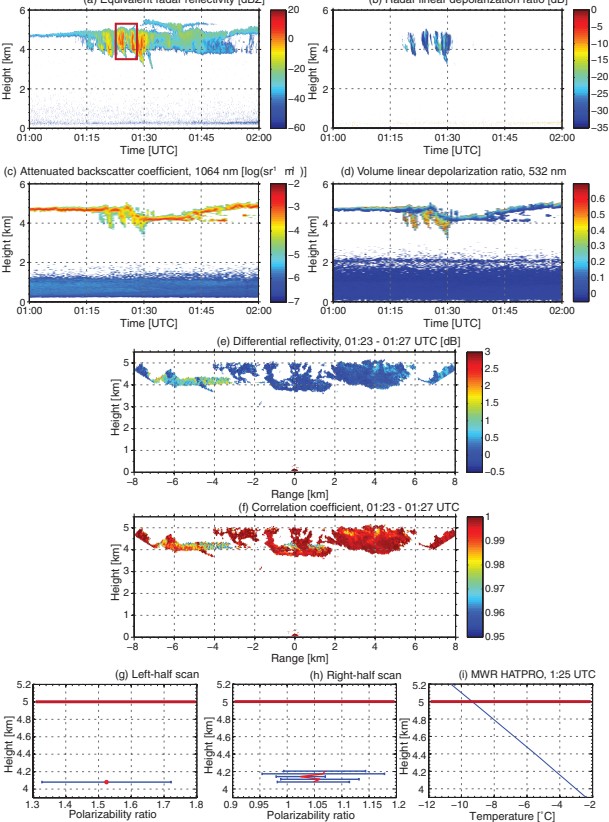

**Figure 3.** Same as in Fig. 2, but for 18 October 2014 and without a photograph.

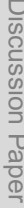

**AMTD**

doi:10.5194/amt-2015-365

**Shape-temperature relationship of pristine ice crystals**

A. Myagkov et al.

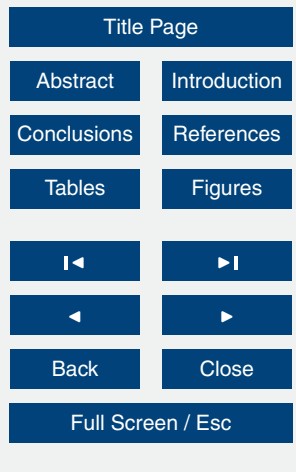

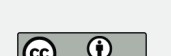

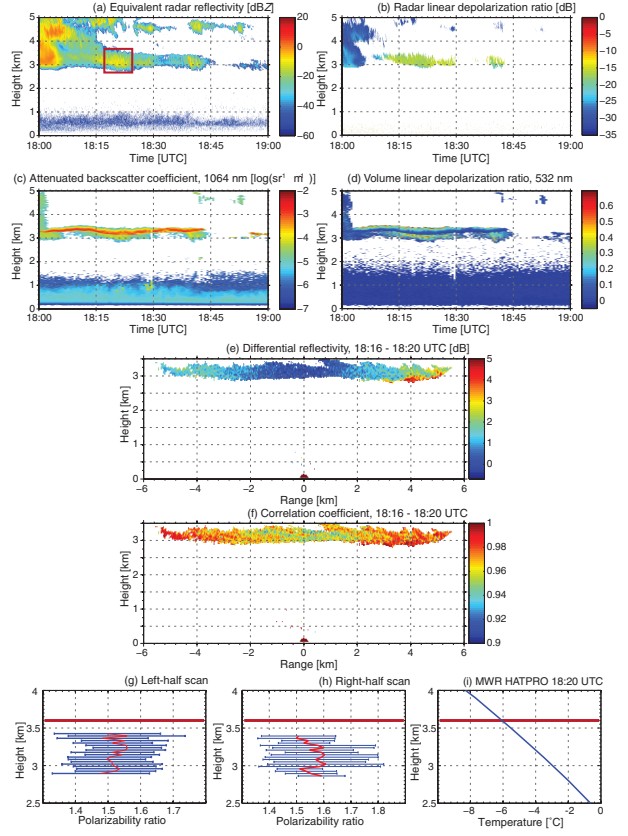

**Figure 4.** Same as in Fig. 2, but for 20 October 2014 and without a photograph.

**AMTD**

doi:10.5194/amt-2015-365

**Shape-temperature relationship of pristine ice crystals**

A. Myagkov et al.

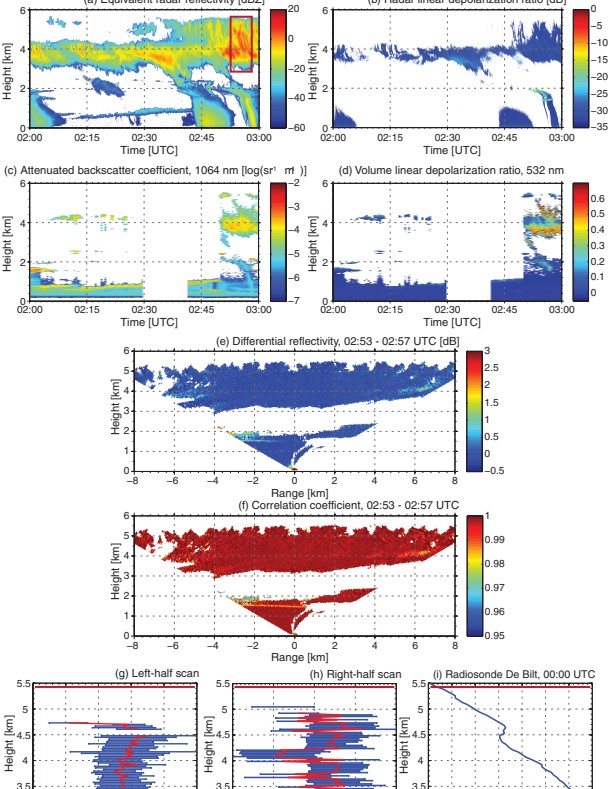

**Figure 5.** Same as in Fig. 2, but for 10 November 2014 and without a photograph.

Discussion Paper | Discussion Paper | Discussion Paper | Discussion Paper |

**AMTD**

doi:10.5194/amt-2015-365

**Shape-temperature relationship of pristine ice crystals**

A. Myagkov et al.

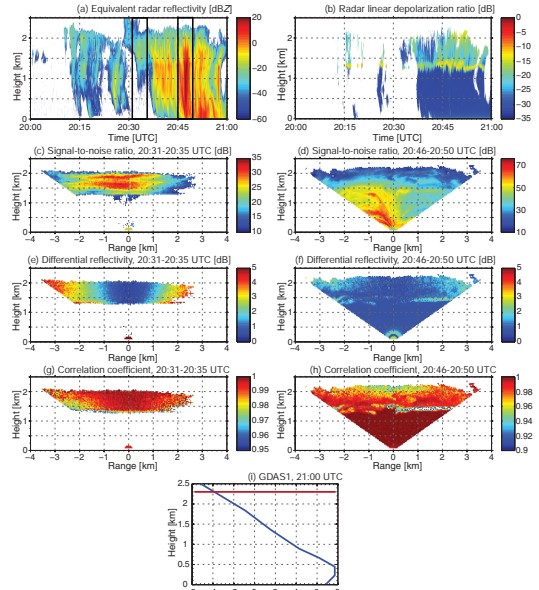

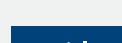
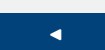
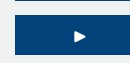
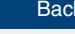
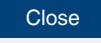

**Figure 6.** Case study of strong short-term variabilty of ice particle shapes observed at Cabauw, the Netherlands, on 7 November 2014. Shown are time-height cross sections of **(a)** equivalent radar reflectivity $Z_e$ from LDR MIRA-35 and **(b)** radar linear depolarization ratio from LDR MIRA-35; range-height cross sections of **(c)** signal-to-noise ratio, **(e)** differential reflectivity $Z_{DR}$ and **(g)** correlation coefficient $\rho_{HV}$ measured by hybrid MIRA-35 from 20:31 to 20:35 UTC; **(d)** signal-to-noise ratio, **(f)** differential reflectivity $Z_{DR}$ and **(h)** correlation coefficient $\rho_{HV}$ measured by hybrid MIRA-35 from 20:46 to 20:50 UTC, **(i)** vertical temperature profile from the GDAS model. Two black rectangles in **(a)** show the analyzed cloud layer and the time periods corresponding to full elevation scans of hybrid MIRA-35. $Z_e$ and LDR were calculated from total powers in the co- and cross-channels. SNR, $Z_{DR}$ and $\rho_{HV}$ were obtained for the spectral peaks. Note, that $Z_{DR}$ and $\rho_{HV}$ are not corrected for the antenna coupling here to make figures more illustrative. Red horizontal lines in **(i)** represents the cloud top at 20:45 UTC.



Discussion Paper | Discussion Paper | Discussion Paper | Discussion Paper | Discussion Paper |

**AMTD**

doi:10.5194/amt-2015-365

**Shape-temperature relationship of pristine ice crystals**

A. Myagkov et al.

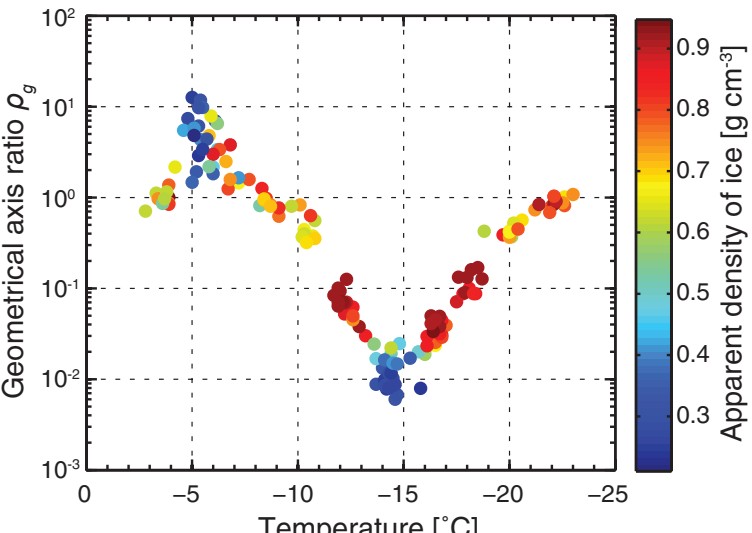

**Figure 7.** Temperature dependence of geometrical axis ratio for particles grown in the free-fall cloud chamber (Takahashi et al., 1991). Apparent density is color-coded. Note, that $\rho_g > 1$ corresponds to prolate particles, $\rho_g < 1$ corresponds to oblate particles. Numerical data were provided by Prof. Takahashi, Hokkaido University of Education, Sapporo, Japan.

Discussion Paper | Discussion Paper | Discussion Paper | Discussion Paper

**AMTD**

doi:10.5194/amt-2015-365

**Shape-temperature relationship of pristine ice crystals**

A. Myagkov et al.

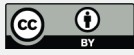

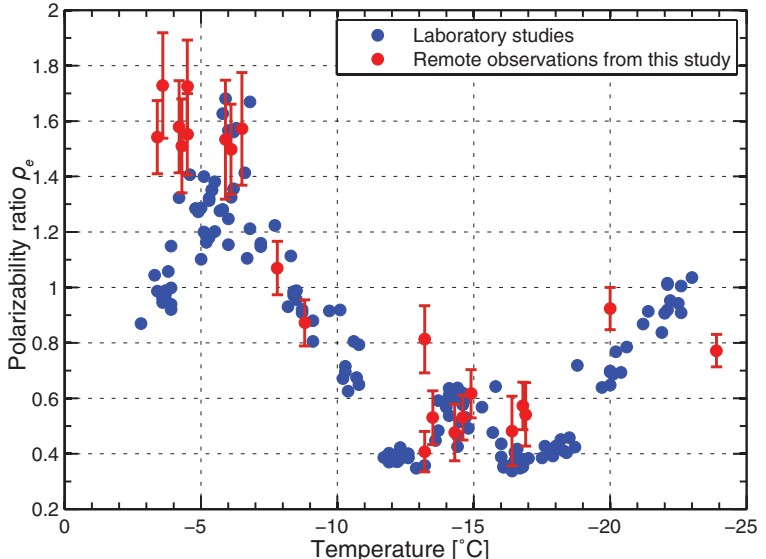

**Figure 8.** Temperature dependence of polarizability ratios for ice crystals grown in the free-fall chamber (blue filled circles) and for ones located close to tops of mixed-phase clouds, retrieved from hybrid MIRA-35 (red filled circles). Note, that $\rho_e > 1$ corresponds to prolate particles, $\rho_e < 1$ corresponds to oblate particles. Vertical red bars represent ±1 standard deviation of observed polarizability ratios. Data from the free-fall chamber (Takahashi et al., 1991) were provided by Prof. Takahashi, Hokkaido University of Education, Sapporo, Japan.