# Peer review of "Shape-temperature relationships of pristine ice crystals derived from polarimetric cloud radar observations during the ACCEPT campaign"

_Atmospheric Measurement Techniques, 2015_

## Referee Comment (RC2) · Anonymous Referee #2 · 14 Apr 2016

General comments:

The paper presents results that documents that measurements of the polarizability ratio (a measure of oblateness vs prolateness dielectric shape) of ice hydrometeors are consistent with general expectations of pristine ice crystal growth at the tops of mixedphase clouds for a ranges of cloud top temperatures spanning -20 C to -3 C. The authors present scattering calculations to show that their polarizability ratio measurements corresponds to laboratory ice grown at similar temperatures at liquid saturation. The data presented in the paper generally supported the results of the paper, so in that respect the paper is worthy of publication.

Specific comments:

My main concern with the paper was with the presentation and conclusions of the paper.

First, the introduction (on mixed-phase clouds) is mostly irrelevant to the topic of the paper. An introduction of the main topic of the paper, polarizability ratios, is completely missing from the introduction, and only appears in the discussion were even then this reader could not grasp what it was. Not being one of the leading experts in the field of polarimetric radar this reviewer certainly was not familiar, other in very general terms, with what polarizability ratio mean in this context, so I was lost from the get go. To check if I was merely ignorant of a new development in field I asked colleagues much more involved in the polarizability ratio. I had to work through the very technical paper (Myagkov et al 2015a) to finally get the discussion I needed. I strongly suggest that the authors replace the current introduction of the paper with a relevant discussion of polarizability ratio to position non-experts to follow the subsequent discussion of the observations. Help the community understand the value of what you are doing.

Second, I would argue that the title of the paper (shape-temperature relationships of pristine ice crystals) does not accurately reflect what is presented. The polarizability ratio is more akin to an "apparent shape" as it depends on the shape and "bulk" density (dielectric properties) of the ice as the authors present. This is clear from Figure 10 in Myankov et al (2015a)which shows that other that being an indicator of oblate/prolate dominated scatterers the ratio does provide much information on the scatterers (i.e. something I can use in my cloud model to characterize ice). Unfortunately the dielectric properties of the ice in general is not know, so what the authors presented in the paper is probably the best one may hope for: the measurements are consistent with the lab data (itself a worthy conclusion). I would suggest a more accurate description of the results it that use of the polarizability ratio (as proposed by Myagkov et al. 2015) is an improvement over previous techniques to identify regions in the cloud dominated by
columnar vs plate-like ice crystals.

I would be great if the authors can contradict my previous two points by showing that we can learn more about the ice found at the top of mixed-phase clouds. Surely that would be a great contribution if they could.

---

## Author Comment (AC1) · 8 May 2016

Please find the responses, introduced changes, and updated versions of the manuscript in the attached zip archive.

Please also note the supplement to this comment: http://www.atmos-meas-tech-discuss.net/amt-2015-365/amt-2015-365-AC1-supplement.zip